# Mesenchymal Stem/Stromal Cells for Therapeutic Angiogenesis

**DOI:** 10.3390/cells12172162

**Published:** 2023-08-28

**Authors:** Farina Mohamad Yusoff, Yukihito Higashi

**Affiliations:** 1Department of Regenerative Medicine, Division of Radiation Medical Science, Research Institute for Radiation Biology and Medicine, Hiroshima University, Hiroshima 734-8553, Japan; drfarinamyusoff@hiroshima-u.ac.jp; 2Division of Regeneration and Medicine, Hiroshima University Hospital, Hiroshima 734-8551, Japan

**Keywords:** critical limb ischemia, mesenchymal stem/stromal cells, therapeutic angiogenesis

## Abstract

Mesenchymal stem/stromal cells (MSCs) are known to possess medicinal properties to facilitate vascular regeneration. Recent advances in the understanding of the utilities of MSCs in physiological/pathological tissue repair and technologies in isolation, expansion, and enhancement strategies have led to the use of MSCs for vascular disease-related treatments. Various conditions, including chronic arterial occlusive disease, diabetic ulcers, and chronic wounds, cause significant morbidity in patients. Therapeutic angiogenesis by cell therapy has led to the possibilities of treatment options in promoting angiogenesis, treating chronic wounds, and improving amputation-free survival. Current perspectives on the options for the use of MSCs for therapeutic angiogenesis in vascular research and in medicine, either as a monotherapy or in combination with conventional interventions, for treating patients with peripheral artery diseases are discussed in this review.

## 1. Introduction

Due to the demographic trends towards an aging population and the projected increase in the incidence of cardiovascular diseases, peripheral artery disease (PAD) is expected to become a larger burden [1]. Critical limb ischemia (CLI), also known as chronic limb-threatening limb ischemia, refers to a condition characterized by chronic (≥2 weeks) ischemic rest pain, nonhealing wounds/ulcers, or gangrene in one or more limbs attributable to objectively proven arterial occlusive disease [2,3,4]. In patients with CLI, treatment involves complex management due to the varying degrees of severity upon presentation with irreversible tissue loss, which can result in significant morbidity and possibly lead to premature mortality. The current limb salvage intervention options have not reduced the number of PAD-related major and minor amputations to the expected optimal levels. Minor amputations are associated with a high risk of major amputation and death. One in 10 patients had an ipsilateral major amputation within the first year after minor amputation, and half of the patients died within 5 years [5,6]. These complications cause significant physical, psychological, and economic burdens for patients and communities [7]. Such progression in patients with chronic ischemia that exceeds the tissue capacity for simple diffusion of oxygen and nutrients in microvasculature territories requires improved treatment strategies in advanced PAD with a focus on restoring the balance for tissue survival through microvascular regeneration using exogenous molecular and cellular agents [8]. To restore blood flow to peripheral ischemic tissues, neovascularization of microvasculature through targeted angiogenesis by cell therapy is investigated.

Mononuclear cells (MNCs), mesenchymal stem/stromal cells (MSCs), or marker-specific subsets of autologous or allogeneic harvested cells with angiogenic properties are examined as potential treatments for PAD patients [9,10,11,12]. MNCs were isolated from bone marrow (BM) or peripheral blood (PB), whereas MSCs can be isolated from a plethora of sources, including bone marrow, peripheral blood, umbilical cord, dental pulp, adipose tissue, and other tissues. Due to the progeniture properties and low immunogenicity of MSCs, as well as technological advancements in harvesting, isolating, and expanding MSCs, these cells are studied for vascular-related medical treatments. The purpose of this review is to discuss the use of MSCs for therapeutic angiogenesis in vascular research and medicine.

## 2. Angiogenesis and Vascular Regeneration

Angiogenesis is the mechanism of formation of new blood vessels that occurs during embryogenic development and in adult individuals [13,14,15,16,17,18,19,20,21,22,23,24,25,26,27,28]. The process of angiogenesis refers to the sprouting of new capillaries from preexisting vessels, whereas the process of arteriogenesis refers to the remodeling of newly formed or preexisting vascular channels into larger, well-muscularized arterioles, and collateral vessels. Angiogenesis, arteriogenesis, and vasculogenesis occur as dynamic processes that are constantly responsive to physiological and pathological stimuli, such as hypoxia, tissue ischemia, inflammation, and shear stress. Multiple factors that stimulate or inhibit vascular regeneration have been identified. Regulation of these factors determines the functional balance in the macro and micro-vasculature and the required responses to repair vascular-related tissue injuries. Table 1 summarizes selected biological factors regulating angiogenesis.

Angiogenesis-related factors are highly responsive to tissue microenvironmental stresses [8,10,12,14,15,20,21,22,23,24,25,26,27,28]. The presence of numerous radical and non-radical molecules leads to stimulation or inhibition of these factors. Autocrine and paracrine controls influence angiogenesis through multiple interactions of various cell types, including monocytes/macrophages, T- and B-cells, mast cells, other inflammatory cells, circulating and resident progenitor cells, vascular endothelial cells, pericytes, and vascular smooth muscle cells. The functions of these cells are hindered in the presence of persistent macro- and micro-vasculature defects.

Microvascular dysfunction in PAD is increasingly appreciated, with evidence for impaired small artery vasoreactivity, decreased nitric oxide signaling, and increased endothelin receptors [29,30]. By promoting effective microvascular regeneration, the reconstitution of blood flow is expected to continuously improve the condition of ischemic tissue regions down their micro-environments that are inaccessible with current interventions.

## 3. Therapeutic Angiogenesis by Cell Therapy

Cell therapy, or cell-based therapy, can be defined as a set of strategies in which live cells with therapeutic purposes are used [12]. The aim of cell therapy is to repair, replace, and/or restore the biological function of a damaged tissue or organ. Thus, the use of stem cells in cell therapy is studied in several areas of cardiovascular medicine [31,32,33,34,35,36,37,38,39,40,41,42,43,44,45,46,47,48,49,50,51,52]. Cells that have been used in cardiovascular studies include BM-MNCs, peripheral blood-derived MNCs, MSCs, adipose-derived stem cells, and circulating endothelial cells. BM-MNCs, a mixed population of single nucleus cells, including monocytes, lymphocytes, and hematopoietic stem and progenitor cells, and MSCs from multiple sources have been studied as therapy options for vascular regeneration (Figure 1). The outcomes of clinical trials on cell therapies have mainly focused on amputation rates, improvement in ulcers/wounds, improvement in rest pain score, ankle-brachial index changes, and improvement in walking distance and mortality. Transcutaneous oxygen tension, formation of collateral vessels, cell doses, route/mode of delivery, and adverse effects have also been investigated in some studies. Some randomized control, non-randomized, and non-controlled clinical trials of cell therapy for PAD are summarized in Table 2.

There have been many clinical trials in which the potential of cell therapy for a large variety of clinical symptoms was examined. As shown in Table 2, cell therapy provided relief for severe PAD symptoms. The beneficial effects of cardiovascular cell-based cell therapy are likely to be mediated by autocrine, paracrine, and possibly endocrine mechanisms [31,32]. Methods for delivery of cells into affected areas, including intraarterial injections, intramuscular implantations, topical delivery using fibrin spray, scaffolds or with collagen as dressings, and injections into wound edges, are investigated. Autologous cell therapy was at the forefront in initiating cellular therapy for patients as it carries fewer risks of cellular/tissue reactions and accelerates the wound healing process by reducing the time needed for host cells to invade the wound tissue with augmented microvasculature regeneration. Allogeneic cell therapy is also studied and may hold a different pathway in determining cellular reactions and tissue suitability.

Therapeutic angiogenesis by cell therapy (TACT) trial aims at improving clinical symptoms in patients with CLI who had no option other than amputation [53]. In various studies mentioned above, positive effects of cell transplantation occurred even without long-term engraftment or survival of transplanted cells, and repeated cell transplantations were also performed. Long-term studies can provide more details on overall clinical outcomes.

## 4. MSCs for Therapeutic Angiogenesis

As one of the most encouraging cell types as a regenerative alternative to conventional interventions, MSCs for therapeutic angiogenesis are investigated as a viable option to overcome the problem of a large amount of BM required for extraction of adequate cells for a therapeutic number of BM-MNCs. Although BM-MNCs have been shown to provide therapeutic benefits in the treatment of CLI, some patients were of an advanced age and had severe complications, such as myocardial ischemia, heart failure, cerebrovascular disease, and renal failure [54,55].

MSCs, also recognized as mesenchymal precursor cells or medicinal signaling cells, comprise a specialized population of progeniture cells that can be differentiated in a laboratory into different types of tissue, and they have been reported to secrete bioactive molecules and facilitate the recovery of ischemic or injured tissues [56,57,58,59,60]. According to recommendations by the Cardiovascular Cell Therapy Research Network, MSCs in culture-expanded cells should be recognized through characteristics of adherence to tissue culture plastic, expressing CD90, CD73, and CD105 markers, not expressing CD45, CD34, CD14, CD11B, CD79a, CD19, or HLA-DR markers, and capacity for multilineage differentiations, such as osteocytes, chondrocytes, and adipocytes. MSCs have been investigated extensively as a potential candidate for PAD cell therapy because of their multi-potency properties, autocrine and paracrine effects, possible transdifferentiation, and immunosuppressive effects in vivo and in vitro [61,62,63,64,65,66,67,68,69,70,71,72,73,74]. Autologous MSCs have been shown to be safe for transplantation, not causing immune rejection, and technically viable for isolation. Allogeneic MSCs have been investigated to offer additional advantages, such as donor selection, availability from various sources, low immunogenicity, and being readily available for use.

### 4.1. Cell Differentiation and/or Transdifferentiation

It has been reported that MSCs are capable of being expanded in vitro and can differentiate into bone, cartilage, muscle, marrow stroma, tendon, fat, and a variety of other connective tissues if placed in permissive cultures [64,65]. Culture-expanded populations are highly selected and fundamentally different from the mixed starting population of stem and progenitor cells that contribute to their generation. Tissue-specific committed progenitors have limited differentiation capacity. When a blood vessel is broken or inflamed, perivascular cells, pericytes, are detached, and some of these liberated cells differentiate into MSCs. Culture-expanded MSCs can be transplanted back into the body, where they are home to sites of injury or inflammation. Those newly homed MSCs are capable of surveying and sensing the microenvironment in which they exist, and they have a programmed response profile of secretory activity for any given microenvironment.

### 4.2. Autocrine and Paracrine Signals

The medicinal properties of exogenous MSCs influence the autocrine and paracrine signaling tissue repair and regeneration predominantly by secreting numerous bioactive factors or the secretome [61,62,63,64,65,66,67,68]. It is thought that multiple protein factors, including hepatocyte growth factor (HGF), fibroblast growth factor, and vascular endothelial growth factor (VEGF), in transplanted MSCs play a major role in angiogenesis in ischemic tissues through autocrine and paracrine signals. The microvessels secrete proangiogenic paracrine factors to induce early blood perfusion. The initiation of the paracrine signaling cascade leads to pericyte detachment, endothelial permeabilization, and endothelial cell migration. Through the enhancement of these effects, vascular regeneration capabilities in ischemic tissue can be enhanced with the transplantation of MSCs.

### 4.3. Anti-Inflammatory and Immunomodulation Effects

MSCs have been found to have anti-inflammatory effects through regulatory T-cell, interleukin-10 (IL-10), and IL-4 functions and to suppress inflammation through functions of T-cells, B-cells, tumor necrosis factor alpha, interferon gamma, IL-12, natural killer cells, and monocytes. On the other hand, toll-like receptor-specific activation of MSCs induces the production of proinflammatory cytokines and chemokines, including IL-6 and IL-8 [69,70,71]. Conditioning of MSCs with proinflammatory factors enhances the immunosuppressive properties of MSCs. These factors influence macrophage polarization in the ischemic tissue microenvironments. Vascular regeneration that occurs after implantation of MSCs into ischemic tissue is the result of concerted anti-inflammatory and immunomodulation effects that reduce tissue damage from overt inflammation and promote tissue regeneration.

### 4.4. Anti-Apoptotic and Anti-Fibrotic Effects

Repair of damaged tissue by transplantation of MSCs possibly occurs through anti-inflammatory, anti-fibrotic, and/or anti-apoptotic effects [72,73]. MSCs interact with surrounding tissues and cells, regulating the extracellular matrix, producing anti-inflammatory molecules through modulating the immune system, preventing cell death, promoting angiogenesis, and possibly playing a dual role in fibrosis development. However, MSCs from different sources have been found to have different effects. MSCs derived from type 1 diabetes mellitus (DM) patients can maintain their normal capability of anti-apoptotic and anti-fibrotic factors, while MSCs from type 2 DM individuals may be dysfunctional with increased rates of senescence and apoptosis and decreased proliferation and angiogenesis potentials. Properties of MSCs can alter and can be altered in pathological and harmful microenvironments. With innate anti-apoptotic and anti-fibrotic components in normal MSCs, enhancement strategies have been applied to harness the optimum benefits from MSCs. The use of autologous or allogeneic MSCs as an anti-fibrotic cell therapy approach has been proposed to reduce fibrosis in different types of tissue [73]. The benefits of MSCs as a potential therapeutic option for specific diseases need to be carefully balanced with their potential risks in clinical settings.

### 4.5. Oxidative Stress

Microvascular dysfunction due to profound and chronic oxidative stress has been the underlying cause of failure in conventional interventions in PAD [74]. Implantation of MSCs that have antioxidant elements can ameliorate microenvironment damage when delivered to injured tissues. Although catalase, superoxide dismutase 1–3, glutathione peroxidase, sirtuin 1, 3, and 6, thioredoxin, and heme oxygenase-1, along with an antioxidant molecule reduced glutathione and redox-sensitive forkhead box O3 signaling, MSCs have been found to be resistant to a certain threshold of overt oxidative stress. Reduction in reactive oxygen species to a beneficial range can mediate recovery from vascular injury and other diseases.

Vascular regeneration requires a series of molecular and cellular interactions that involve the spatial distribution and temporal expression of substantial signaling and matrix molecules within the angiogenic microenvironment. This complex and fine regulation of vascular growth should be considered in designing effective biology-oriented therapeutic strategies for the treatment of ischemic diseases. MNC and MSC transplantation enhances neovascularization by supplying multiple angiogenic factors, such as VEGF, bFGF, and angiopoietin-1. The implanted cells enhance the mobilization of endothelial cells to the ischemic injury sites and participate in the organization of vascular structures. Iwase et al. [62] reported that injection of equal numbers of MNCs or MSCs into ischemic muscle to compare the therapeutic effects of the two types of cells and showed that MSC transplantation markedly increased blood perfusion and capillary density in the ischemic hindlimb compared with the effects of MNC transplantation. Perfusion recovery with transplantation of 1 × 10^6^ MSCs was equivalent to that with transplantation of 5 × 10^6^ MNCs. Compared with MNCs, MSCs survived well in an ischemic environment. Experimental models of ischemic tissues have demonstrated neovascularization capabilities using MSCs from diverse sources, and autologous and allogeneic MSCs have their own advantages and disadvantages [73]. There has been bidirectional interaction between these implanted cells and tissue microenvironments. Implantation of autologous MSCs improved endothelium-dependent vasodilation [61,63]. One of the possible mechanisms by which MSC implantation augments endothelium-dependent vasodilation is by increasing shear stress resulting from blood flow. MSCs are known for their immunomodulatory properties and allogeneic MSCs can be safely administered without causing a significant immune response. In addition to the induction of angiogenesis in the ischemic limb, MSC implantation augments endothelium-dependent vasodilation through an increase in nitric oxide production. Therapeutic angiogenesis by using MSCs as cell therapy to treat limb ischemia with remedial vascular regeneration can be achieved by synchronizing the MSC properties in cellular differentiation/transdifferentiation, autocrine and paracrine signaling, and anti-inflammatory and immunomodulatory effects along with the presence of antioxidants and anti-fibrotic and anti-apoptotic elements.

Figure 2 summarizes therapeutic angiogenesis by cell therapy for vascular diseases. As shown in Table 1, BM is used as a source of MNCs and MSCs. However, cells from peripheral blood, umbilical cord/placenta, and adipose tissue have also been used in clinical trials. Isolation and expansion of MSCs from multiple sources (Figure 1) are needed to obtain a therapeutic number of MSCs for providing adequate treatment to patients.

## 5. Other Developments in MSCs

Modified MSCs for therapeutic purposes would need to be specifically identified, recognized, and documented. Diversity in techniques to harvest, process, expand, and deliver MSCs with therapeutic properties have been investigated by various methods, including sequestration of exosomes or micro-vesicles, application of a conditioned medium, hypoxia induction in cell culture, use of mechanical exposure, transfection with specific genes in combination with scaffolds or biomaterials. This area of regenerative medicine is expanded by translating the results of fundamental science research into sophisticated, personalized, and applicable clinical practices.

### 5.1. Hypoxia-Induced MSCs

MSCs might have some resistance to a certain range of oxygen limitation [75,76]. The microenvironment of MSCs in tissue depots is characterized by a considerably low oxygen (O_2_) partial pressure. Hypoxia activates many stress and survival pathways in MSCs. Hypoxia-induced factor 1-alpha (HIF-1α) influences the colony-forming mesenchymal progenitors to promote self-renewal of the population of MSCs. Specific forms of HIF-1 have been shown to promote vascular regeneration through arterial destabilization, increased vascular permeability, extracellular matrix remodeling, migration and proliferation of endothelial cells, endothelial cell sprouting, tube formation, and cell-to-cell contact, recruitment of and interaction with pericytes, and maintenance of vessel integrity [77,78,79]. The ex vivo exposure time for hypoxia-induced MSCs in studies varied from 0 to 72 h with an oxygen concentration ranging from 0% to 5%. Prolonged hypoxia and inflammation cause maladaptation of HIF-1, which can lead to ineffective VEGF regulation and impair microvascular regeneration and tissue recovery. The proficiency of MSCs for vascular regeneration and tissue healing can be enhanced by hypoxia preconditioning when cultured in a 1% O_2_ environment for 24 h. Yusoff et al. [78] reported that the pretreatment of MSCs with a hypoxia condition and implantation of hypoxia-induced MSCs can advance neovascularization capability with enhanced therapeutic angiogenic effects that can improve limb perfusion in critical limb ischemia.

### 5.2. Mechanically Induced MSCs

MSCs are considered to be one of the most promising populations of cells for the development of vascular tissue engineering [80,81,82,83]. Efforts have been made to differentiate MSCs towards vascular cell phenotypes not only by manipulating biochemical factors but also by applying hemodynamic forces, such as shear stress and cyclic strain. Experiments showed that these factors were able to facilitate the differentiation of MSCs into ECs or vascular smooth muscle cells. Low-intensity pulsed ultrasound (LIPUS) with a frequency of 1–3 MHz and an intensity of <1 W/cm^2^ is also investigated to determine the potential of ultrasound stimulation for differentiation of MSCs. LIPUS was found to influence the MEK-ERK signaling pathway [12,84,85,86].

### 5.3. Exosomes/Microvesicles in MSCs

The bioactive molecules in MSCs are contained in exosomes and microvesicles. These components of MSCs can be isolated and purified, and they are characterized by sizes between 50–200 nm and specific expression of exosome-associated markers [87,88,89]. The exosomes are secreted by paracrine cells and play a key role in tissue repair and regeneration. They promote angiogenesis and upregulate the early inflammatory responses. Exosomes can be derived from different types of MSCs, and exosomes from different sources have unique characteristics. As signaling molecules, exosomes in MSCs not only exert the same effects as those of MSCs but also have a more stable membrane structure than those of MSCs. Exosomes can be developed to integrate into providing the properties of MSCs in clinical practice. Exosomes can be integrally developed to provide MSC characteristics in the clinical setting.

### 5.4. Scaffolds and Biomaterials

Scaffolds and biomaterials were designed for accommodating and harnessing the properties of MSCs to address different needs in tissue regeneration [90,91,92]. The properties of MSCs were incorporated into scaffolds and biomaterials, such as microbubbles or magnetic particles, to facilitate the attachment of cells to the desired locations. Along with targeted tissue delivery of MSCs, the application of scaffolds and biomaterials provides additional support for tissue regeneration, especially in larger nonhealing wound areas and specific sites of interest.

These strategies were developed to acquire and optimize the effects of MSCs in a variety of disease conditions. Cells can be delivered as monotherapy, with or without repeated implantations, and cell-based treatments can be further improved with the enhanced properties from the induced effects of MSCs.

## 6. Challenges in the Use of MSCs for Therapeutic Angiogenesis

Clinical studies have been conducted worldwide to investigate the safety and effectiveness of cell therapy, and numerous reviews have been published [9,10,11,12,31,93,94,95,96,97,98,99,100]. A vascular therapeutic strategy is adopted for inducing angiogenesis to stimulate neo-vascularization in the restoration of blood circulation in the affected limb for the treatment of PAD. The concerted mechanisms of vascular regeneration would require multi-levels and multi-prong processes to accommodate the extensive network of blood vessels in the body. The expected outcomes are not only an improvement in blood flow in the microvasculature but also a reduction in pain and an improvement in wound healing, amputation-free survival, and quality of life.

Conclusions based on the results of meta-analysis regarding cell therapy for CLI have been inconsistent. In systemic reviews, indiscriminate pooled data with heterogeneous backgrounds from some studies were used. The presence of non-responders in those studies has not been ascertained, and patients with less advanced PAD seem to be more responsive to regenerative strategies. Indeed, the risk of selection bias in primary or secondary studies and an inadequate number of trials discussed in reviews can lead to uncertainty in drawing a definite conclusion. It is not justifiable to surmise the overall ineffectiveness of a specific therapeutic method based on data from clinical trials using a wide spectrum of approaches. Large-scale trials have not yet been carried out.

Autologous MSCs require a few weeks for isolation, in vitro expansion, and release, and patient-derived cells may be affected by age, underlying diseases, such as diabetes mellitus, and tobacco exposure, which may also decrease the number and function of therapeutic cells [10]. Systemic and local metabolic states of biological sources can affect the composition, prevalence, and properties of the cell population. Allogeneic MSCs may cause immune rejection and may have donor-donor heterogeneity, specific immunological memory, and potential disease candidate genes. As noted above, there are several challenges regarding the use of autologous or allogeneic MSCs. Other factors that need to be overcome are as follows: (1) Isolation and banking of MSCs: Techniques of isolation of cells are being continuously revised. These techniques require professional training with verified identification and handling techniques. MSCs can be banked after cell culture, and cell implantation can be repeatedly performed. (2) Homing: Cell homing occurs as signaling attracts cells to injured tissue. Various methods to utilize the properties of cells for tissue regeneration have been investigated. However, some methods for the delivery of cells may lead to the inadequacy of therapeutic effects. Homing can be unsuccessful as the cells are unable to reach the site of affected tissue for the intended treatment. (3) Cell dose: In addition to the possibility that homing may not be effective due to uneven delivery of cells into the damaged tissue, the number of cells implanted also varied in previous studies. The optimal number of cells, or cell dose, for effective and consistent results has not been investigated in detail. (4) Cell implantation, technical skills, and patient care: Cell implantations should be performed at specialized centers for cell therapy to adequately address technical issues. Specialized technical skills in cell implantations should be optimized to provide adequate cell transplantations for patients. Patient care should be adequately reviewed by trained professionals in vascular medicine and/or regenerative medicine for patients with PAD.

Historically, there has been heterogeneity in the description of the composition of stem cells. Tissue-specific stem and progenitor cells can be found in small amounts in adult tissues and possess self-renewal capacity. The ambiguity has been misused by certain quarters in misleading advertisements and marketing for self-deserving profits that have caused setbacks to actual scientific progression. Clinical and experimental studies are ongoing to use MSCs more efficiently and safely for therapeutic purposes.

## 7. Summary and Perspectives

MSCs for therapeutic angiogenesis by cell therapy can be considered a possible treatment option to improve clinical symptoms and amputation-free survival in patients with severe and chronic PAD. Thus, strategies to consolidate efforts towards using MSCs for therapeutic angiogenesis need to be established. Strategic considerations include (1) selection of MSC type should be based on the underlying disease and the intended targets and (2) appropriate and suitable candidates with vascular diseases, such as PAD, CLI, and ischemic heart disease, must be identified. Patients with diabetes, chronic kidney disease, and other comorbidities may require additional assessments to determine their suitability for cell therapy. (3) The method for cell delivery must be considered. The method for delivering cells is a critical consideration. Depending on the location of the disease and the nature of the treatment, targeted delivery methods may be appropriate. Further studies on single and/or repeated cell transplantation with or without current vascular interventional treatments should be carried out to improve clinical outcomes and patients’ quality of life. (4) The response should be monitored. One important aspect of cell therapy is monitoring the patient’s response. Regular follow-up visits should be conducted to assess the patient’s clinical status and laboratory parameters and perform imaging tests. Adequate engagements with patients, caregivers, and multidisciplinary teams that are involved in the short-term and long-term care of the patients would optimize treatment outcomes. (5) Standardized safety protocols must be adhered to. Safety is a paramount concern. It is important to take appropriate safety measures, including screening patients for potential adverse effects and monitoring them for any complications following treatments. Treatments should only be performed at centers that specialize in cell therapy.

Approximately one-third of patients with CLI who have undergone conventional interventions still require leg amputation within three years [29]. The TACT trial has been conducted for over 15 years to provide options for patients with severe PAD who have no option in conventional interventions and for whom cell therapy has the potential to modify the natural history of intractable CLI [100]. In recent years, evidence of the effectiveness of cell therapy using MSCs for the treatment of CLI cases has been obtained. The presence of non-responders in cell therapy needs to be addressed. Only therapeutic strategies with a combination of different types of cells in ischemic tissue have so far been used. Treatment strategies in combination with conventional interventions have not been explored. Rapid and sustained recovery from an ischemic limb and chronic wound would improve the overall quality of life. It is hoped that more medical institutes will use and continue to improve TACT trial methods as treatment options for patients with intractable PAD.

## Figures and Tables

**Figure 1 cells-12-02162-f001:**
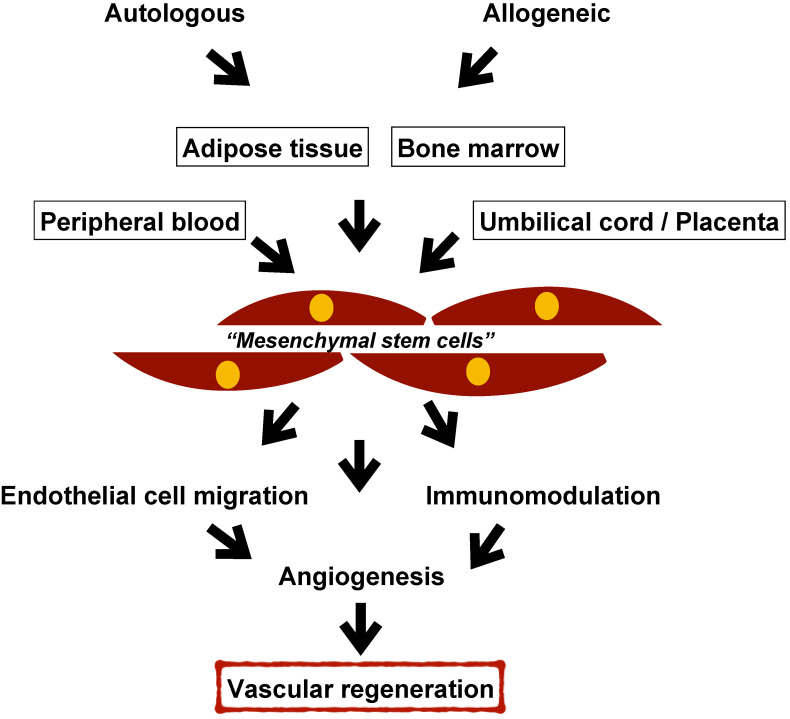
Mesenchymal stem/stromal cells from multiple sources for vascular regeneration. (Figure is designed by authors using Microsoft Office 365, 2019).

**Figure 2 cells-12-02162-f002:**
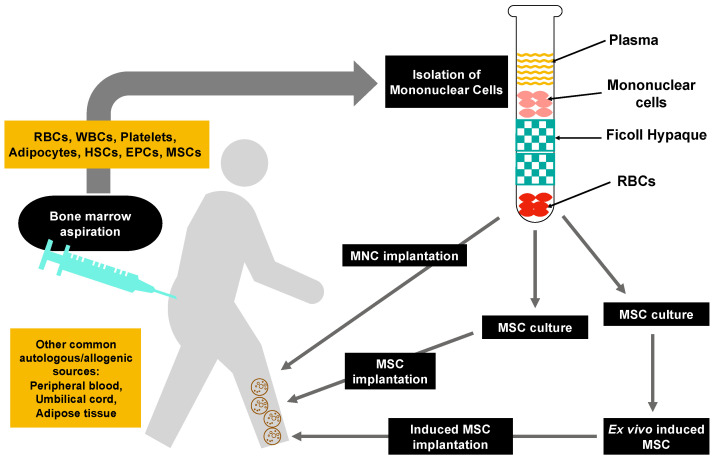
Therapeutic angiogenesis by cell therapy for vascular diseases. EPC indicates endothelial progenitor cell; HSC, hematopoietic stem cell; MSC, mesenchymal stem/stromal cell; RBC, red blood cell; WBC, white blood cell. (Figure is designed by authors using Microsoft Office 365, 2019).

**Table 1 cells-12-02162-t001:** Biological factors regulating angiogenesis.

Factors	Mechanisms
Angiopoietin-1	Endothelial cell chemotaxis, prevention of excessive vascular permeability, formation of lumens, and stabilization of vessels via endothelial cell-mural cell interactions
Angiopoietin-2	Vessel destabilization, detachment of VSMCs, and degradation of extracellular matrix
EGF	Promotion of vascular endothelial cell growth
FGF	Induction of angiogenesis, endothelial cell proliferation, lumen formation, recruitment of inflammatory cells, pericytes and VSMCs, and vessel maturation
HGF	Acts as a multi-functional cytokine on cells of mainly epithelial origin. Due to its ability to stimulate mitogenesis, cell motility, and matrix invasion, it has a chief role in angiogenesis and tissue regeneration
HIF-1α	Upregulation of several genes under low-oxygen conditions, including glycolysis enzymes, and VEGF
IGF	A key regulator of cellular proliferation and differentiation, reduces apoptosis and collagen deposition, enhances angiogenesis
ILs	IL-6, IL-8, and IL-10 act as chemoattractants and are involved in modulation of inflammatory cells
NO	Vasodilation, co-factor for VEGFs, FGFs, and other angiogenic factors
MMP	Degradation of the extracellular matrix, activation of angiogenesis-inducting factors
PDGF	Involved in migration of vascular endothelial cells. Promotes the recruitment of perivascular cells to support nascent vessels
PGC-1α	One of the important regulators of oxidative metabolism and mitochondrial function and is also involved in induction of VEGF
VEGFs	Involved in angiogenesis and lymph-angiogenesis, endothelial cell migration and proliferation, and neo-vascularization and induces vascular permeability

EGF indicates epidermal growth factor; FGF, fibroblast growth factor; HGF, hepatocyte growth factor; HIF-1 α, hypoxia-induced factor 1-alpha; IGF, insulin-like growth factor; ILs, interleukins; MMP, matrix metalloproteinases; NO, nitric oxide; PDGF, platelet-derived growth factor; PGC-1α, peroxisome proliferator-activated receptor gamma coactivator 1-alpha; VEGFs, vascular endothelial growth factors.

**Table 2 cells-12-02162-t002:** Selected randomized, non-randomized, and non-controlled clinical trials of cell therapy for peripheral artery disease.

Cell Type	Indication	Method of Cell Delivery	Cell Number	Number of Subjects	Outcomes	Reference
**BM-MNCs**						
BM-MNCs (including CD34^+^ cells)	CLI with Rutherford class 4–6	Intramuscular	1.6 × 10^9^	7	Improvements in TcPO2, pain-free walking time, acetylcholine-mediated endothelium-dependent blood flow	Higashi et al., 2004 [33]
BM-MNCs	ASO/TAO with Rutherford class 4–6	Intramuscular	N/A	115	Improvements in pain scale, ulcer size, and walking distance, reduced amputation rates	Matoba et al., 2008 [34]
BM-MNCs	No optional CLI	Intra-arterial, intra-muscular	N/A	27	Improvements in ABI, pain score, pain-free walking distance	Van Tongeren et al., 2008 [35]
BM-MNCs	CLI with Rutherford class 4–6	Intramuscular	1.1–3.0 × 10^9^	51	Improvements in RC and walking distance, reduced analgesics consumption	Amman et al., 2009 [36]
BM-MNCs	Advanced severe chronic limb ischemia	Intramuscular	10 × 10^8^	15	Improvements in ABI and ulcer healing	Zafarghandi et al., 2010 [37]
BM-MNCs (including CD34+ cells)	CLI with Rutherford class 4–6	Intra-arterial	1 × 10^8^	40	Improvements in ulcer healing and rest pain with cell therapy No significant improvement found in ABI or limb salvage	Walter et al., 2011 [38]
BM-MNCs	TAO with CLI	Intramuscular	0.5 × 10^9^	22	Improvements in SPP, pain score, and TcPO2	Fujioka et al., 2023 [39]
**BM-MNCs/BM-MSCs**						
BM-MNCs BM-MSCs	Type 2 DM with CLI	Intramuscular	9.6 × 10^8^ BM-MNCs/9.3 × 10^8^ BM-MSCs	41	Improvements in pain-free walking time and wound healing	Lu et al., 2011 [40]
BM-MNCs BM-MSCs	Limb ischemia	Intramuscular	9 × 10^8^ BM-MNCs and 9 × 10^6^ BM-MSCs or 1.8 × 10^9^ BM-MNCs and 1.8 × 10^7^ BM-MSCs		Improvements in walking time, ankle-brachial index, significant increase of perfusion in the treated limbs compared with the respective control legs	Lasala et al., 2012 [41]
BM-MNCs BM-MSCs (including CD34^+^ cells)	Type 2 DM with CLI	Intramuscular	9.3 ± 1.1 × 10^8^ BM-MSCs or 9.6 ± 1.1 × 10^8^ BM-MNCs	41	Improvements in ulcerative healing and reduction in ulcer recurrence, limb salvage	Lu et al., 2019 [42]
**BM-MSCs**						
BM-MSCs	Diabetic or non-diabetic, failed or not suitable for revascularization	Intramuscular	2 × 10^6^ cells/kg body weight	20	Improvements in rest pain, TcPO2, ABI and ulcer healing	Gupta et al., 2013 [43]
BM-MSCs	TAO who had not responded to or were not eligible for revascularization	Intramuscular	1 and 2 × 10^6^ cells/kg body weight	72	Reduction in rest pain, improvement in healing of ulcers, improvements in ABI and total walking distance No significant difference in the number of collateral vessels and amputation-free survival	Gupta et al., 2017 [44]
BM-MSCs	CLI with required amputation	Intramuscular	5 × 10^6^ per injection	66	Improvements in mortality, limb status, changes in pain score	Wijnand et al., 2018 [45]
**PB-MNCs**						
PB-MNCs (including CD34^+^ cells)	5 TAO, and 1 ASO	Intramuscular	3.9 × 10^10^	6	Improvements in ABI, ischemic ulcer, walking distance	Ishida et al., 2005 [46]
PB-MNCs	Rutherford class 4–6	Intramuscular	1 × 10^7^	40	Improvements in ABI and pain scores	Ozturk et al., 2011 [47]
PB-MNCs	DM with CLI	Intramuscular	1 × 10^7^	21	Improvements in ABI and amputation rates	Mohammadzadeh et al., 2013 [48]
**PB-MNCs/BM-MNCs**						
PB-MNCs (including CD34+ cells) BM-MNCs (including CD34+ cells)	ASO with Rutherford class 1–6	Intramuscular	1 × 10^9^ PB-MNCs /1 × 10^8^ BM-MNCs	150	Patients who received PB-MNCs showed improved ABI and rest pain compared with those in patients who received BM-MNCs	Huang et al., 2007 [49]
**Others**						
UCB-MSCs	No-option patients with end-stage CLTI	Intramuscular	1 × 10^7^	8	Improvement of ulcer healing	Yang et al. [50]
ATMSCs	TAO with diabetic foot	Intramuscular	3 × 10^8^	12	Improvements in walking distance, pain rating scale, and clinical symptoms	Lee et al., 2012 [51]
ADSCs	CLI (Fontaine class III–IV) with no other option for standard revascularization	Intramuscular	6.9 × 10^7^	29	Improvement in major amputation- free survival rates in no-option CLI patients	Shimizu et al., 2022 [52]

ABI indicates ankle brachial index; ADSCs, adipose-derived stem/stromal cells; ATMSCs, adipose tissue-derived mesenchymal stem/stromal cells; ASO, arteriosclerosis obliterans; BM-MNCs, bone marrow–derived mononuclear cells; BM-MSCs, bone marrow–derived mesenchymal stem/stromal cells; CD, cluster of differentiation; CLI, critical limb ischemia; DM, diabetes mellitus, N/A, not available, PAD, peripheral arterial disease, PB-MNCs, peripheral blood–derived mononuclear cells; TAO, thromboangiitis obliterans; TcPO2, transcutaneous oxygen pressure and UCB-MSCs, umbilical cord blood mesenchymal stem cells.

## Data Availability

Not applicable.

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
