# Peer review of "Mesenchymal Stem/Stromal Cells for Therapeutic Angiogenesis"

_cells, 2023, doi:10.3390/cells12172162_

Round 1

Reviewer 1 Report

The manuscript entitled " Mesenchymal stem cells for therapeutic angiogenesis" is an interesting title and of current importance.

The manuscript title does not match with the literature cited. An overall general properties of MSCs are given with very little focus over angiogenesis. The literature over MSCs angiogenesis is too less to be able to conclude anything out of the manuscript.

The writing is also poor appearing as if little importance over proof -read has been given.

I doubt if the manuscript shall be accepted in the present form and with such a limited literature against the given title.

Poor writing appearing as if no proof-reading has been done.

Author Response

Point 1: The manuscript entitled "Mesenchymal stem cells for therapeutic angiogenesis" is an interesting title and of current importance.

The manuscript title does not match with the literature cited. An overall general properties of MSCs are given with very little focus over angiogenesis. The literature over MSCs angiogenesis is too less to be able to conclude anything out of the manuscript.

Response 1: We appreciate the reviewer’s comments on the title of manuscript and its concept of current importance. The concept of the manuscript is to discuss the development of therapeutic angiogenesis by cell therapy in chronological order from the use of MNCs to the use of MSCs in later years for vascular regeneration. Mesenchymal stem/stromal cells are known to possess medicinal properties to facilitate vascular regeneration. Recent advances in the understanding of the utilities of MSCs in physiological/pathological tissue repair and technologies in isolation, expansion and enhancement strategies have led to the use of MSCs for vascular disease-related treatments. Various conditions including chronic arterial occlusive disease, diabetic ulcers and chronic wounds cause significant morbidity in patients. Therapeutic angiogenesis by cell therapy has led to the possibilities of treatment options in promoting angiogenesis, treating chronic wounds, and improving amputation-free survival. Current perspectives on the options for the use of MSCs for therapeutic angiogenesis in vascular research and in medicine, either as a monotherapy or in combination with conventional interventions, for treating patients with peripheral artery diseases are discussed in this review. Discussion of angiogenesis induced by MSCs has been expanded in the revised manuscript. Vascular regeneration requires a series of molecular and cellular interactions that involve the spatial distribution and temporal expression of substantial signaling and matrix molecules within the angiogenic microenvironment. This complex and fine regulation of vascular growth should be considered in designing effective biology-oriented therapeutic strategies for the treatment of ischemic diseases. MNC and MSC transplantation enhances neovascularization by supplying multiple angiogenic factors such as VEGF, bFGF, and angiopoietin-1. The implanted cells enhance mobilization of endothelial cells to the ischemic injury sites and participate in organization of vascular structures. Iwase et al. reported that injection of equal numbers of MNCs and MSCs into ischemic muscle to compare the therapeutic effects of the two types of cells and showed that MSC transplantation markedly increased blood perfusion and capillary density in the ischemic hindlimb compared with the effects of MNC transplantation. Perfusion recovery with transplantation of 1x106 MSCs was equivalent to that with transplatation of 5x106MNCs. Compared with MNCs, MSCs survived well in an ischemic environment. Experimental models of ischemic tissues have demonstrated neovascularization capabilities using MSCs from diverse sources, and autologous and allogeneic MSCs have their own advantages and disadvantages. There has been bidirectional interaction between these implanted cells and tissue microenvironments. Implantation of autologous MSCs improved endothelium-dependent vasodilation. One of the possible mechanisms by which MSC implantation augments endothelium-dependent vasodilation is by increasing shear stress resulting from blood flow. MSCs are known for their immunomodulatory properties and allogeneic MSCs can be safely administered without causing a significant immune response. In addition to induction of angiogenesis in the ischemic limb, MSC implantation augments of endothelium-dependent vasodilation through an increase in nitric oxide production. Therapeutic angiogenesis by using MSCs as cell therapy to treat limb ischemia with remedial vascular regeneration can be achieved by synchronizing the MSC properties in cellular differentiation/transdifferentiation, autocrine and paracrine signaling, and anti-inflammatory and immunomodulatory effects along with the presence of antioxidants and antifibrotic and antiapoptotic elements. These statements have been added to Section 4 (page 9, lines 224-252).  

Point 2: The writing is also poor appearing as if little importance over proof -read has been given.

Response 2: We value the reviewer’s feedback on the writing and we have made substantial corrections to the style of writing in the revised manuscript. As pointed out by the reviewer, the process of angiogenesis refers to the sprouting of new capillaries from preexisting vessels, whereas the process of arteriogenesis refers to the remodeling of newly formed or preexisting vascular channels into larger, well-muscularized arterioles and collateral vessels. Angiogenesis, arteriogenesis, and vasculogenesis occur as dynamic processes that are constantly responsive to physiological and pathological stimuli such as hypoxia, tissue ischemia, inflammation, and shear stress. These revised statements have been added to Section 2 (page 2, lines 55-61). The repeated words of ‘have been identified’, in page 2, line 59 in the previous manuscript, have been deleted in the revised manuscript. In page 7, line 127 of the previous manuscript, the words ‘by cell therapy’ were deleted in the revised manuscript. The statement of ‘MSC were reported to possess the ability to be expanded in vitro and the ability to differentiate into bone, cartilage, muscle, marrow stroma, tendon, fat and a variety of other connective tissues in permissive culture’ in the previous manuscript has been corrected to ‘It has been reported that MSCs are capable of being expanded in vitro and can differentiate into bone, cartilage, muscle, marrow stroma, tendon, fat, and a variety of other connective tissues if placed in permissive cultures’ in page 8, lines 158-160 in the revised manuscript. All revised statements were written in red.

Point 3: I doubt if the manuscript shall be accepted in the present form and with such a limited literature against the given title.

Response 3: We appreciate the reviewer’s opinion on the previous form of the manuscript and the advice given on literature against the given title. The appropriate comments from the reviewer have enabled us to improve the manuscript. As previously mentioned, the concept of the manuscript is to discuss the development of therapeutic angiogenesis by cell therapy in chronological order from the use MNCs to the use of MSCs in later years for vascular regeneration. In the revised manuscript, the discussion and literature on MSCs and angiogenesis have been expanded.

Reviewer 2 Report

The review presented is of interest and well articulated but needs major reviews to be published in this journal.

As the authors will know, the term stem has now been abandoned and the term stromal is currently used. Then correct in the paper.

The bibliography is scarce and there are entire paragraphs without  references. Are these assumptions or data of the authors? also in this case a bibliographic reference must be made explicit.

Table 2 must be sorted by placing all data relating to the same type of cells consecutively. It would be clearer to report a table for each cell line considered.

Were the figures created by the authors? with what program? Write it in the caption together with the license or report the bibliographic reference.

English needs to be reviewed and corrected

Author Response

Point 1: The review presented is of interest and well articulated but needs major reviews to be published in this journal.

Response 1: We appreciate the reviewer’s comments that the review presented is of interest and well articulated. We appreciate the opportunity to revise and improve the manuscript.

Point 2: As the authors will know, the term stem has now been abandoned and the term stromal is currently used. Then correct in the paper.

Response 2: We agree with the reviewer’s recommendation to use the term stromal in the paper. We have restructured the term MSC to represent Mesenchymal Stem/Stromal Cells in the revised version.  

Point 3: The bibliography is scarce and there are entire paragraphs without  references. Are these assumptions or data of the authors? also in this case a bibliographic reference must be made explicit.

Response 3: We appreciate the reviewer’s advice on the bibliographic references. Statements in the manuscript are constructed in accordance to references in the beginning of the paragraph. We have added explicit bibliographic references in the revised manuscript as advised.

Point 4: Table 2 must be sorted by placing all data relating to the same type of cells consecutively. It would be clearer to report a table for each cell line considered.

Response 4: We appriciate the reviewer’s recommendation to sort Table 2 by placing all data relating to the same type of cells consercutively. The previous version that was presented in chronological order has been revised into data presentation in accordance to the type of cells in pages 4-7 in the updated manuscript.

Point 5: Were the figures created by the authors? with what program? Write it in the caption together with the license or report the bibliographic reference.

Response 5: We appriciate the reviewer’s advice concerning the figures. The figures were designed by the authors using Microsoft Office 365, 2019. This has been ststed in the legends for Figures 1 and 2.

Reviewer 3 Report

The paper is well written, comprehensible and well referenced with latest publications. As a review of course we cannot pretend to add some news on the field but it is always good to have a feed-back o the previous works and this do a good job.

Author Response

Point 1: The paper is well written, comprehensible and well referenced with latest publications. As a review of course we cannot pretend to add some news on the field but it is always good to have a feed-back o the previous works and this do a good job.

Response 1: We appreciate the reviewer’s encouraging comments on the paper. We agree with the reviewer that it is always good to have a feedback on the previous works and to continue improving the protocols towards achieving better outcomes for the patients. We have revised the previous paper to correct some details and add discussion on the potential of MSCs for therapeutic angiogenesis. We are hopeful that our paper will ignite more interest among physicians and scientists in this topic.

Round 2

Reviewer 2 Report

No other comments